# Directional Message Passing on Molecular Graphs via Synthetic Coordinates

**Johannes Gasteiger, Chandan Yeshwanth, Stephan Günnemann**
Technical University of Munich, Germany
`{j.gasteiger,yeshwant,guennemann}@in.tum.de`

## Abstract

Graph neural networks that leverage coordinates via directional message passing have recently set the state of the art on multiple molecular property prediction tasks. However, they rely on atom position information that is often unavailable, and obtaining it is usually prohibitively expensive or even impossible. In this paper we propose synthetic coordinates that enable the use of advanced GNNs without requiring the true molecular configuration. We propose two distances as synthetic coordinates: Distance bounds that specify the rough range of molecular configurations, and graph-based distances using a symmetric variant of personalized PageRank. To leverage both distance and angular information we propose a method of transforming normal graph neural networks into directional MPNNs. We show that with this transformation we can reduce the error of a normal graph neural network by 55 % on the ZINC benchmark. We furthermore set the state of the art on ZINC and coordinate-free QM9 by incorporating synthetic coordinates in the SMP and DimeNet$^{++}$ models. Our implementation is available online. [1]

## 1   Introduction

Graph neural networks (GNNs) have set the state of the art on many tasks of molecular machine learning, such as the prediction of quantum mechanical properties (Gilmer et al., 2017), solubility (Wu et al., 2018), or the generation of new molecules (Jin et al., 2020). Thanks to their fast inference time, good generalization and scalability, GNNs are thus promising to revolutionize large parts of chemistry, from ab-initio quantum mechanical simulations and reaction kinetics to synthesis planning and drug discovery. Atom positions are central to many of these tasks, but unavailable in most cases. Many tasks in chemistry instead use a more coarse-grained representation: The molecular graph. Unfortunately, this representation makes many predictive tasks substantially harder, and GNNs have performed significantly better when they have access to the exact molecular configuration (Gilmer et al., 2017). Missing atom positions furthermore preclude the use of many advanced GNNs that were developed with coordinates in mind.

In this work we aim to fill in this information with well-defined coordinates constructed purely from the molecular graph. Regular approximation methods for generating atom positions often do not benefit model performance, due to the fundamental ambiguity of molecular configurations. The energy landscape of molecules can have multiple local minima, and a molecule can be in any of multiple different minima, known as conformers. In this work, we propose to circumvent this problem by incorporating the conformational ambiguity via empirical distance *bounds*. Instead of yielding a potentially wrong configuration, these bounds only estimate the range of viable molecular geometries. They are thus valid regardless of which state the molecule was in when generating the data.

---

[1] `https://www.daml.in.tum.de/synthetic-coordinates`

35th Conference on Neural Information Processing Systems (NeurIPS 2021).

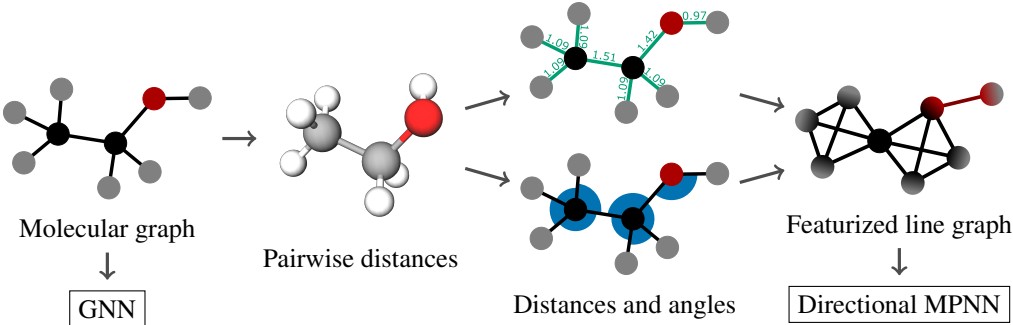

Figure 1: Illustration of transforming a regular molecular graph (ethanol) to a line graph with synthetic coordinates. We first calculate all (bounds of) pairwise distances using our synthetic coordinates. We then calculate the (bounds of) distances and angles for the molecular graph. Finally, we convert the molecular graph to its line graph and embed the distances and angles as features. This process allows us to convert a regular GNN to a directional MPNN, which improves its accuracy and allows to incorporate angular information.

A molecular configuration is fully specified by the pairwise distances between all atoms, due to rotational, translational, and reflectional invariance. We can thus obtain a molecular geometry from any method that provides pairwise distances between atoms. Since directional message passing does not require the full molecular geometry, these distances do not need to correspond to an actual three-dimensional configuration. We leverage this generality and propose purely graph-based distances calculated from a symmetric variant of personalized PageRank (PPR) as a second set of coordinates. This distance performs surprisingly well, despite incorporating no chemical knowledge. Both the distance bounds and the symmetric PPR distance require no hand-tuning and can be calculated efficiently, even for large molecules.

We leverage these two variants of *synthetic coordinates* to transform regular GNNs into directional message passing, as illustrated in Fig. 1. We first calculate the synthetic, pairwise distances for the given molecular graph. Based on these, we calculate the edge distances and angles between edges. Finally, we compute the molecule's line graph. Executing a GNN on the line graph improves its expressivity (Garg et al., 2020) and allows us to incorporate angular information. We use the original node and edge attributes together with the distances as node attributes, and the obtained angles as edge attributes. The GNN is then executed on this featurized line graph instead of the original graph. Our experiments show this transformation can significantly improve the performance of the underlying GNN, across multiple models and datasets. Incorporating synthetic coordinates reduces the error of a normal GNN by 55 %, putting it on par with the current state of the art. Our enhanced version of the SMP model (Vignac et al., 2020) improves upon the current state of the art on ZINC by 21 %, and DimeNet$^{++}$ (Gasteiger et al., 2020a) with synthetic coordinates outcompetes previous methods on coordinate-free QM9 by 20 %. In summary, our core contributions are:

- Well-defined synthetic coordinates based on node distances and simple molecular bounds, which significantly improve the performance of GNNs for molecules.
- A general scheme of converting a normal GNN into a directional MPNN, which can improve performance and allows incorporating both distance and angular information.

## 2 Directional message passing

**Graph neural networks.** To use GNNs for molecules we represent them as graphs $\mathcal{G} = (\mathcal{V}, \mathcal{E})$, where the atoms define the node set $\mathcal{V}$ and the interactions the edge set $\mathcal{E}$. These interactions are usually the bonds of the molecular graph, but they can also be all atoms pairs within a given cutoff of e.g. 5 Å. In this work we focus on an extension of message passing neural networks (MPNNs) (Gilmer et al., 2017). MPNNs embed each atom $u$ separately as $\boldsymbol{h}_u \in \mathbb{R}^H$, and can additionally use interaction embeddings $\boldsymbol{e}_{(uv)} \in \mathbb{R}^{H_e}$. These embeddings are updated in each layer using messages passed between neighboring nodes, starting with the atom features $\boldsymbol{h}_u^{(0)} = \boldsymbol{x}_u^{(\mathcal{V})}$ (e.g. its type) and the interaction features $\boldsymbol{e}_{(uv)}^{(0)} = \boldsymbol{x}_{(uv)}^{(\mathcal{E})}$ (e.g. the bond type or a distance representation). Extended

MPNNs can be expressed via the following two equations:

$$\boldsymbol{h}_u^{(l+1)} = f_{\text{update}}(\boldsymbol{h}_u^{(l)}, \underset{v \in \mathcal{N}_u}{\text{Agg}}\, [f_{\text{msg}}(\boldsymbol{h}_u^{(l)}, \boldsymbol{h}_v^{(l)}, \boldsymbol{e}_{(uv)}^{(l)})]), \tag{1}$$

$$\boldsymbol{e}_{(uv)}^{(l+1)} = f_{\text{edge}}(\boldsymbol{h}_u^{(l+1)}, \boldsymbol{h}_v^{(l+1)}, \boldsymbol{e}_{(uv)}^{(l)}). \tag{2}$$

The atom and interaction update functions $f_{\text{node}}$ and $f_{\text{edge}}$ and the message function $f_{\text{msg}}$ are learnable functions, such as simple linear layers or arbitrarily complex neural networks. The aggregation Agg over the atom's neighbors $\mathcal{N}_u$ is usually a simple summation.

**Line graph.** The directed line graph $\mathcal{L}(\mathcal{G}) = (\mathcal{V}_\mathcal{L}, \mathcal{E}_\mathcal{L})$ expresses the adjacencies between the directed edges in $\mathcal{G}$. Its nodes are the directed edges of the original graph $\mathcal{V}_\mathcal{L} = \{(u, v) \mid u \in \mathcal{V}, v \in \mathcal{N}_u\}$. For undirected graphs like molecular graphs, every undirected edge $\{u, v\}$ is split into two directed edges $(u, v)$ and $(v, u)$. Two nodes in $\mathcal{L}(\mathcal{G})$ are connected if the corresponding edges in $\mathcal{G}$ share a node, i.e. $\mathcal{E}_\mathcal{L} = \{((u, v), (v, w)) \mid (u, v), (v, w) \in \mathcal{V}_\mathcal{L}\}$. We obtain node features for the line graph by embedding the original node and edge features as $\boldsymbol{x}_{(uv)}^{(\mathcal{V}_\mathcal{L})} = f_{\text{emb}}(\boldsymbol{x}_u^{(\mathcal{V})}, \boldsymbol{x}_v^{(\mathcal{V})}, \boldsymbol{x}_{(uv)}^{(\mathcal{E})})$. The line graph can furthermore incorporate additional features for atom triplets as edge features $\boldsymbol{x}_{(uvw)}^{(\mathcal{E}_\mathcal{L})}$, such as the angle between bonds or interactions.

**Directional message passing.** Directional MPNNs improve upon regular MPNNs in two ways. First, they embed the directed messages instead of the nodes in the graph, essentially operating on the directed line graph. Models using only this first step are also known as directed MPNNs or line graph neural networks (Dai et al., 2016; Yang et al., 2019; Chen et al., 2019). Directed MPNNs are strictly more expressive than regular MPNNs (Morris et al., 2020). We can transform any MPNN to a directed MPNN simply by executing it on the directed line graph instead of the original graph.
Second, for graphs with nodes that are embedded in an inner product space (such as molecules in 3D space) the directed edges correspond to directions in that space, via $\boldsymbol{x}_{(uv)}^{(\mathcal{V}_\mathcal{L})} = \boldsymbol{x}_u^{(\mathcal{V})} - \boldsymbol{x}_v^{(\mathcal{V})}$. Directional MPNNs leverage this connection to better represent the molecular configuration, usually by using the angles in $\boldsymbol{x}_{(uvw)}^{(\mathcal{E}_\mathcal{L})}$ (Gasteiger et al., 2020b). To fully leverage both aspects of directional MPNNs we therefore need some form of coordinates.

**Expressivity of GNNs.** A central limitation of GNNs is their inability of distinguishing between certain non-isomorphic graphs. For example, GNNs are not able to distinguish between a hexagon and two triangles if all nodes and edges have the same features. More specifically, Xu et al. (2019); Morris et al. (2019) have shown that GNNs are only as powerful as the 1-Weisfeiler-Lehman (WL) test of isomorphism. While it is still possible to construct indistinguishable examples for directional MPNNs, this is significantly more difficult (Garg et al., 2020). Dym & Maron (2021) have shown that MPNNs using SO(3) group representations and atom positions are even universal, i.e. able to approximate any continuous function to arbitrary precision. This demonstrates that coordinates can alleviate and even solve this central limitation of GNNs.

## 3 Molecular configurations

To prevent any pitfalls when constructing synthetic coordinates for GNNs based on chemical knowledge we first need to consider the properties of atomic positions in a molecule and how they are obtained. At first glance these positions might seem like an obvious and straightforward property. However, molecular configurations are actually ambiguous and difficult to obtain, even for small molecules. This misconception has even led some works to suggest semi-supervised learning methods leveraging positions, effectively treating them as abundant input features (Hao et al., 2020). To clarify this issue we will next describe the complexity behind molecular configurations and how to approximate them efficiently.

**Finding molecular configurations.** The atoms of a molecule can in principle be at any arbitrary position. However, most of these configurations will lead to an extremely high energy and are thus very unlikely to be observed in nature. A molecular configuration thus usually refers to the atom positions at or close to equilibrium, i.e. at the molecule's energy minimum. To find these positions we have to search the molecule's energy landscape and solve a non-convex optimization problem. This is in fact a bilevel optimization problem, where the atom positions are optimized in the outer and the electron wavefunctions in the inner task. These wave functions can then be ignored in the outer task;

they only influence the energy and the forces $\boldsymbol{F}_i = -\frac{\partial E}{\partial \boldsymbol{x}_i}$ acting on each atomic nucleus. We can then use these forces for gradient-based optimization, and avoid saddle points by using quasi-Newton methods.

**Difficulties.** The above optimization process is very expensive due to the quantum mechanical (QM) computations required for optimizing the electron wavefunction at each gradient step. It is orders of magnitude more expensive than calculating the energy of a *given* molecular configuration, since we need to calculate the energy's gradient for each optimization step. Furthermore, the optimization will only converge to a local, and not the global minimum. And in fact, the global minimum is not the only state of interest — *any* reasonably low local minimum of the energy landscape is a valid configuration, known as a conformer. A molecule thus does not have a unique configuration; it can be in any of these states. Their statistical distribution and the interaction between them is central for many molecular properties. This ambiguity of atom positions poses a fundamental limit on how precise molecular predictions can be without knowing the exact (ensemble of) configurations. For example, without knowing the molecule's conformer we can not reasonably predict its energy at a precision below roughly 60 meV (Grimme, 2019) — except for small, rigid molecules that do not have multiple conformers (e.g. benzene).

**Approximating energies and forces.** The most prominent way of accelerating the process of finding a valid molecular configuration is by approximating its most expensive part: The quantum mechanical optimization of the electron wavefunction. There is a large hierarchy of methods with various runtime versus accuracy trade-offs (Folmsbee & Hutchison, 2021). The cheapest class of methods are force fields. They allow running molecular dynamics simulations with millions of atoms, and can estimate the equilibrium structure of a small molecule in less than one second. Force fields approximate the quantum-mechanical interactions via a closed-form, differentiable function that only depends on the atom positions. One common example is the Merck Molecular Force Field (MMFF94) (Halgren, 1996). MMFF94 calculates the molecular energy based on interatomic distances, angles, dihedral angles, and long-range interaction terms. Each term is approximated using an analytic equation with empirically chosen coefficients that depend on the involved atom types. Forces are obtained via the analytical gradients $\boldsymbol{F}_i = -\frac{\partial E}{\partial \boldsymbol{x}_i}$, and conformers via gradient-based optimization. Generating configurations with force fields is fast enough to even generate a large ensemble of conformers. However, the resulting conformers are highly biased and require corrections based on expensive QM-based methods for reasonably approximating the molecule's true distribution (Ebejer et al., 2012; Kanal et al., 2018).

**Directly predicting the configuration.** There are multiple methods that circumvent the optimization process to quickly generate low-energy conformers for a given molecular graph. Distance geometry methods generate conformers using an experimental database of ideal bond lengths, bond angles, and torsional angles (Havel, 2002). The ETKDG method combines this with empirical torsional angle preferences (Riniker & Landrum, 2015). Multiple machine learning methods for generating conformers have also recently been proposed (Weinreich et al., 2021; Lemm et al., 2021).

**Restrictions for ML.** All of the above methods yield reasonable molecular configurations. However, they often require many initializations and a considerable amount of hand-tuning to yield a good result for every molecule in a dataset. Furthermore, the obtained conformer might not even be the correct one for the data of interest. The data could have been generated by a different conformer or by a statistical ensemble of multiple conformers. The configuration of a wrong conformer can cause our model to overfit to the false training data and cause bad generalization (see Sec. 6). To solve this issue we could try to generate an ensemble of conformers and embed their distribution. However, cheap generation methods yield strongly biased ensembles and would thus require expensive post-processing, defeating the purpose of fast and scalable machine learning (ML) methods (Ebejer et al., 2012; Kanal et al., 2018). We propose to instead solve this issue by using less precise *synthetic* coordinates that are easier and cheaper to obtain.

## 4    Synthetic coordinates

**Molecular distance bounds.** To circumvent the issues associated with conformational ambiguity, we propose to use pairwise distance *bounds* instead of simple coordinates, i.e. minimum and maximum distances $d_{(\min)}$ and $d_{(\max)}$ for every pair of atoms. These bounds only provide the chemical information we are certain of, without being falsely accurate. Specifically, we use the distance bounds

provided by RDKit (RDKit, 2021). These bounds provide different estimates depending on how the atoms are bonded in the molecular graph. The edges in the molecular graph correspond to directly bonding atoms, whose bounds are calculated as the equilibrium distance (as parametrized in the universal force field (UFF) (Rappe et al., 1992)) plus or minus a tolerance of 0.01 Å. The angles between triplets of atoms are estimated based on bond hybridization and whether an atom is part of a ring. The distance bounds between two-hop neighbors are then calculated based on this angle, the bond length, and a tolerance of 0.04 Å, or 0.08 Å for atoms larger than Aluminium. Pairwise distances between higher-order neighbors are not relevant for our method, since we only use the distances and angles of the molecular graph. The distance bounds are then refined using the triangle inequality. Note that these bounds depend almost exclusively on the directly involved atoms. They thus only provide local structural information.

Based on these distance bounds we calculate three different angles for directional MPNNs: The maximally and minimally realizable angles, and the center angle. We obtain them using standard trigonometry, via

$$\alpha_{ijk}^{(a)} = \arccos\left(\frac{d_{(b),ij}^2 + d_{(b),jk}^2 - d_{(a),ik}^2}{2d_{(b),ij}d_{(b),jk}}\right), \tag{3}$$

where (a) = (max) and (b) = (min) for the maximally realizable angle, (a) = (min) and (b) = (max) for the minimally realizable angle, and (a) = (b) = (center) for the center angle, with the center distance $d_{(center)} = (d_{(min)} + d_{(max)})/2$. These distance and angle bounds hold for all reasonable molecular structures and thus provide valuable, general information for our model. Their calculation requires no hand-tuning, takes only a few milliseconds, and worked out-of-the-box for every molecule we investigated.

**Graph-based distances.** Directional MPNNs only use the distances of interactions and the angles between interactions; they do not require a full three-dimensional geometry. We leverage this generality to propose a second distance based on a common graph-based proximity measure: Personalized PageRank (PPR) (Page et al., 1998), also known as random walks with restart. PPR measures how close two atoms in the molecular graph are by calculating the probability that a random walker starting at atom $i$ ends up at atom $j$. At each step, the random walker jumps to any neighbor of the current atom with equal probability, and teleports back to the original atom $i$ with probability $\alpha$. To satisfy the symmetry property of a metric we use a variant of PPR that uses the symmetrically normalized transition matrix, i.e.

$$\mathbf{\Pi}^{\text{sppr}} = \alpha(\mathbf{I}_N - (1-\alpha)\mathbf{D}^{-1/2}\mathbf{A}\mathbf{D}^{-1/2})^{-1}, \tag{4}$$

with the teleport probability $\alpha \in (0,1]$, the adjacency matrix $\mathbf{A}$, and the diagonal degree matrix $\mathbf{D}_{ij} = \sum_k \mathbf{A}_{ik}\delta_{ij}$. We found that this method works well even without considering any bond type information in $\mathbf{A}$. We convert $\mathbf{\Pi}^{\text{sppr}}$ to a distance via

$$d_{\text{sppr},ij} = \sqrt{\mathbf{\Pi}_{ii}^{\text{sppr}} + \mathbf{\Pi}_{jj}^{\text{sppr}} - 2\mathbf{\Pi}_{ij}^{\text{sppr}}}. \tag{5}$$

Note that $\mathbf{\Pi}^{\text{sppr}}$ defines a positive definite kernel, and this is the induced distance in its reproducing kernel Hilbert space. It therefore satisfies all properties of a metric, i.e. identity of indiscernibles, symmetry, and the triangle inequality (Berg et al., 1984, Chapter 3, §3). However, $d_{\text{sppr},ij}$ is a general metric and does *not* yield atom positions in 3D. This is a purely graph-based measure that does not incorporate any chemical knowledge. It reflects how central an atom is in the molecular graph, and how important another atom is to this one, based on the overall network of bonds. It thus only helps the GNN better reflect and process the molecular graph structure. Fig. 2 shows an example of $d_{\text{sppr}}$ on ethanol. Since the law of cosines holds for any inner product space we can calculate the angles for directional message passing via

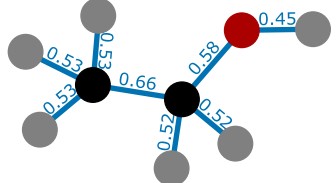

Figure 2: $d_{\text{sppr}}$ distance between direct neighbors on ethanol.

$$\alpha_{ijk} = \arccos\left(\frac{d_{ij}^2 + d_{jk}^2 - d_{ik}^2}{2d_{ij}d_{jk}}\right). \tag{6}$$

Note that the bounds- and graph-based distances encode orthogonal information. The former is solely based on the global molecular graph structure, while the latter provides purely local chemical knowledge. Instead of just choosing one or the other we can therefore combine both to obtain the benefits of both.

**Representing distances and angles.** The additional structural information can directly be incorporated into existing models as edge features. For this purpose, we propose to first represent the distances using $N_{\text{RBF}}$ Gaussian radial basis functions (RBF), i.e.

$$h_{\text{RBF},n}(d_{ij}) = \exp^{-1/2(d_{ij}-c_n)^2/\sigma^2}, \tag{7}$$

where the Gaussian centers $c_n$ are set uniformly between 0 and the overall maximum distance, $n \in [0, N_{\text{RBF}}]$, and $\sigma = c_1 - c_0$ is set as the distance between two neighboring centers. The angles are similarly represented using $N_{\text{ABF}}$ cosine angular basis functions (ABF), i.e.

$$h_{\text{ABF},n}(\alpha_{ijk}) = \cos(n\alpha_{ijk}), \tag{8}$$

with $n \in [0, N_{\text{ABF}}]$. We then transform these features using two linear layers. The first layer is global and uses a small output dimension to force the model to learn a well-generalizing intermediate representation. The second layer is specific to each GNN layer, enabling more flexibility. Overall, we obtain the distance-based edge features $\boldsymbol{e}_{ij}$ and angle-based triplet features $\boldsymbol{a}_{ijk}$ in layer $l$ via

$$\boldsymbol{e}_{ij}^{(l)} = \boldsymbol{W}_{\text{RBF2}}^{(l)} \boldsymbol{W}_{\text{RBF1}}(\boldsymbol{h}_{\text{RBF}}(d_{ij}) \| \boldsymbol{x}_{ij}^{(\mathcal{E})}), \tag{9}$$

$$\boldsymbol{a}_{ijk}^{(l)} = \boldsymbol{W}_{\text{ABF2}}^{(l)} \boldsymbol{W}_{\text{ABF1}} \boldsymbol{h}_{\text{ABF}}(\alpha_{ijk}), \tag{10}$$

where $\boldsymbol{W}_{\text{RBF2}}^{(l)}$ and $\boldsymbol{W}_{\text{ABF2}}^{(l)}$ are layer-wise learned weight matrices, $\boldsymbol{W}_{\text{RBF1}}$ and $\boldsymbol{W}_{\text{ABF1}}$ are global learned weight matrices, $\|$ denotes concatenation, and $\boldsymbol{x}_{ij}^{(\mathcal{E})}$ are bond (edge) features. We can furthermore combine multiple synthetic coordinates by concatenating their representations $\boldsymbol{h}_{\text{RBF}}$ and $\boldsymbol{h}_{\text{ABF}}$. Note that for DimeNet$^{++}$ we use the original basis transformation instead of the one described here.

# 5 Related work

**Graph neural networks.** Sperduti & Starita (1997); Baskin et al. (1997) proposed the first models resembling modern GNNs. Gori et al. (2005); Scarselli et al. (2009) were the first to use the name GNN, but these models are quite different to current GNNs, as described in Sec. 2. GNNs became widely adopted after their potential in a wide range of graph-related tasks was shown by Kipf & Welling (2017); Veličković et al. (2018); Gasteiger et al. (2019); Defferrard et al. (2016); Bruna et al. (2014). Notably, Beaini et al. (2021) use the Laplacian eigenvectors of a graph to enable anisotropic aggregation in MPNNs. This approach is related to our synthetic coordinates. However, it is not rotationally invariant w.r.t. the directions induced by the eigenvectors, and unsuited for enabling existing directional MPNNs.

**GNNs for molecules.** Molecules have always played a central role in the development of GNNs, both for the very first GNNs (Baskin et al., 1997) and during the modern era of GNNs (Duvenaud et al., 2015; Gilmer et al., 2017). GNNs have been particularly successful when leveraging coordinates (Schütt et al., 2017; Unke & Meuwly, 2019), but many variants only rely on the molecular graph (Fey et al., 2020).

**Directionality in GNNs.** Incorporating directionality in molecular MPNNs is currently a very active and successful area of research. These methods can roughly be divided into two classes: Models based on SO(3) group representations (Thomas et al., 2018; Anderson et al., 2019), and models incorporating directional information directly (Gasteiger et al., 2020b). Multiple promising models have recently been proposed for both classes (Fuchs et al., 2020; Batzner et al., 2021; Schütt et al., 2021; Liu et al., 2021; Satorras et al., 2021). While we focus on directional message passing in this paper, all of these methods can benefit from synthetic coordinates.

**Molecular representations.** Molecular fingerprints are a useful tool for comparing molecules, e.g. for machine learning. Popular examples include extended connectivity fingerprints (ECFP), also known as Morgan or circular fingerprints (Rogers & Hahn, 2010), MACCS keys (Durant et al., 2002), MHFP (Probst & Reymond, 2018), the subgraph-based RDKit fingerprint (RDKit, 2021), and

the SELFIES string representation (Krenn et al., 2020). These can be viewed as an alternative or supplement to synthetic coordinates. However, unlike synthetic coordinates they do not leverage the peculiarities of directional MPNNs, and can usually only be used with regressors that are not graph-based. Another class of molecular representations aims at better encoding the geometry of a molecule (Faber et al., 2017). Examples include FCHL (Faber et al., 2018; Christensen et al., 2020), smooth overlap of atomic positions (SOAP) (Bartók et al., 2013), and atomic spectrum of London and ATM potential (aSLATM) (Huang & von Lilienfeld, 2020). OrbNet is an example of a GNN that enhances its input with such a representation (Qiao et al., 2020). Obviously, none of these can be used without access to the molecular configuration.

## 6    Experiments

### 6.1    Experimental setup

We use three common benchmarks to evaluate the proposed synthetic coordinates: Coordinate-free QM9 (Ramakrishnan et al., 2014, CC0 license), ZINC (Irwin et al., 2012), and ogbg-molhiv (Hu et al., 2020, MIT license). QM9 contains various quantum mechanical properties of equilibrium conformers of small molecules with up to nine heavy atoms. To exclude effects from regular chemical information we use all available edge (bond types) and node features (acceptor/donor, aromaticity, hybridization). However, unlike previous work we do not use the Mulliken partial charges. These are computed by quantum mechanical calculations that use the molecule's configuration. They thus lead to information leakage and defeat the purpose of QM9's regression task. We use the same data split as Brockschmidt (2020) for QM9, i.e. 10 000 molecules for the validation and test sets, and the remaining ∼110 000 molecules for training. Note that the properties in QM9 fundamentally depend on the molecular configuration. The predictions in coordinate-free QM9 should thus be viewed as estimates for the equilibrium configurations. There are fundamental limits to the accuracy achievable in this setup, as discussed in Sec. 3. The goal in ZINC is to predict the penalized logP (also called "constrained solubility" in some works), given by $y = \text{logP} - \text{SAS} - \text{cycles}$ (Jin et al., 2018), where $\text{logP}$ is the water-octanol partition coefficient, $\text{SAS}$ is the synthetic accessibility score (Ertl & Schuffenhauer, 2009), and $\text{cycles}$ denotes the number of cycles with more than six atoms. Penalized logP is a score commonly used for training molecular generation models (Kusner et al., 2017). We use 10 000 training, 1000 validation, and 1000 test molecules, as established by Dwivedi et al. (2020) and provided by PyTorch Geometric (Fey & Lenssen, 2019). For ogbg-molhiv we need to predict whether a molecule inhibits HIV virus replication. It contains 41 127 graphs, out of which 80 % are training samples, and 10 % each are validation and test samples, as provided by the official data splits. We report the mean and standard deviation across five runs for ZINC and ogbg-molhiv. Due to computational constraints we only report single results on QM9. The experiments were run on GPUs using an internal cluster equipped mainly with NVIDIA GeForce GTX 1080Ti.

We aim to answer two questions with our experiments: 1. Do synthetic coordinates improve the performance of existing GNNs? 2. Does transforming existing GNNs to directional MPNNs improve accuracy? To answer these questions we investigate three GNNs: DeeperGCN (Li et al., 2020, MIT license), structural message passing (SMP) (Vignac et al., 2020, MIT license), and DimeNet$^{++}$ (Gasteiger et al., 2020a, Hippocratic license). We show step-by-step how the changes affect the resulting error of each model. For easier comparison we also provide published results of other state-of-the-art GNNs: Gated GCN, MPNN-JT (Fey et al., 2020), GIN (Xu et al., 2019; Fey et al., 2020), PNA (Corso et al., 2020), DGN (Beaini et al., 2021), SMP (Vignac et al., 2020), GNN-FiLM (Brockschmidt, 2020), and GNN-FiLM+FA (Alon & Yahav, 2021).

### 6.2    Model hyperparameters

To prevent overfitting we use the SMP and DimeNet$^{++}$ models and hyperparameters largely as-is, without any further optimization. Similarly, we chose the DeeperGCN variant and hyperparameters based on the ogbg-molhiv dataset, and did not further tune on ZINC. More specifically, we use the DeeperGCN (Li et al., 2020) with 12 ResGCN+ blocks, mean aggregation in the graph convolution, and average pooling to obtain the graph embedding. For SMP (Vignac et al., 2020) we use 12 layers, 8 towers, an internal representation of size 32 and no residual connections. For both DeeperGCN and SMP we use an embedding size of 256, and distance and angle bases of size 16 and 18, respectively, with a bottleneck dimension of 4 between the global basis embedding and the local embedding in

Table 1: Ablation study for transforming DeeperGCN and SMP to directional MPNNs (MAE on ZINC). Every step improves the error of DeeperGCN, resulting in a 55 % improvement. The combined bounds+PPR encoding performs best. *Replicated using the reference implementation.

| | | SMP | DeeperGCN |
|---|---|---|---|
| Basic | | $0.159 \pm 0.028$* | $0.317 \pm 0.021$ |
| +distance | Bounds | $0.124 \pm 0.002$ | $0.264 \pm 0.003$ |
| | PPR | $0.151 \pm 0.008$ | $0.227 \pm 0.006$ |
| | Bounds+PPR | $0.121 \pm 0.006$ | $0.228 \pm 0.005$ |
| +distance & line graph | Bounds | $\mathbf{0.112 \pm 0.004}$ | $0.212 \pm 0.008$ |
| | PPR | $0.150 \pm 0.003$ | $0.194 \pm 0.009$ |
| +distance, line graph & angle | Bounds | $\mathbf{0.113 \pm 0.003}$ | $0.180 \pm 0.007$ |
| | PPR | $0.153 \pm 0.005$ | $0.158 \pm 0.005$ |
| | Bounds+PPR | $\mathbf{0.109 \pm 0.004}$ | $\mathbf{0.142 \pm 0.006}$ |

Table 2: MAE on ZINC. SMP with synthetic coordinates outcompetes previous models by 21 %, without any hyperparameter tuning.

| Model | MAE |
|---|---|
| Gated GCN | 0.282 |
| GIN | 0.252 |
| PNA | 0.188 |
| DGN | 0.168 |
| MPNN-JT | 0.151 |
| SMP | 0.138 |
| DeeperGCN-SC | $0.142 \pm 0.006$ |
| SMP-SC | $\mathbf{0.109 \pm 0.004}$ |

Table 3: Comparison of different distance generation methods for DeeperGCN on ZINC (MAE). Our simpler, faster, and more principled methods (bounds, PPR) perform better than more sophisticated conformer generation methods.

| | MMFF94 | ETKDG | Bounds | PPR | Bounds+PPR |
|---|---|---|---|---|---|
| Distance | $0.324 \pm 0.012$ | $0.329 \pm 0.022$ | $0.264 \pm 0.003$ | $\mathbf{0.227 \pm 0.006}$ | $\mathbf{0.228 \pm 0.005}$ |
| Distance & line graph | $0.232 \pm 0.008$ | $0.234 \pm 0.007$ | $0.212 \pm 0.008$ | $0.194 \pm 0.009$ | $\mathbf{0.178 \pm 0.009}$ |
| Distance, line graph & angle | $0.236 \pm 0.011$ | $0.274 \pm 0.012$ | $0.180 \pm 0.007$ | $0.158 \pm 0.005$ | $\mathbf{0.142 \pm 0.006}$ |

each layer. We train all models on ZINC with the same training hyperparameters as SMP, particularly the same learning rate schedule with a patience of 100 and minimum learning rate of $1 \times 10^{-5}$.

For DimeNet$^{++}$ we use a cutoff of 2.5 Å, radial and spherical bases of size 12, embedding size 128, output embedding size 256, basis embedding size 8 and 4 blocks. We use the same optimization parameters - learning rate 0.001, 3000 warmup steps and a decay rate of 0.01.

### 6.3 Results

**Transforming existing GNNs.** Table 1 shows that DeeperGCN's errors improve for each step of the transformation: Adding distance information, switching to the line graph, and adding angles. Interestingly, the PPR distance reduces the error more than molecular distance bounds do. This suggests that this structural information is more relevant for the GNN than the rough bounds. SMP benefits less from using the line graph and angles. This is likely due to SMP already encoding structural information as part of its architecture. Using both the PPR distance and molecular distance bounds improves the performance further for both models. Table 3 shows that using more expensive methods of generating conformers yields a higher error than our simple and fast methods. As discussed in Sec. 3, this can be attributed to the ambiguities of different molecular conformers. DeeperGCN with synthetic coordinates performs similarly well to the best models proposed previously, while the enhanced SMP sets a new state of the art on this dataset, as shown in Table 2.

**Enhancing DimeNet$^{++}$.** DimeNet was originally developed for molecular dynamics and other use cases that provide the true atom positions, such as the full QM9 dataset. Despite this, we can still use it as-is without available positional information by setting the used distance and angle embeddings to constants. DimeNet$^{++}$ still performs surprisingly well in this form, as shown in Table 4. However, its performance increases significantly if we provide it with the proposed synthetic coordinates. Notably, the PPR distance again causes a larger improvement than the molecular distance bounds. Combining both distances still performs best, though. Table 5 furthermore shows that DimeNet$^{++}$ sets the state of the art for coordinate-less QM9 on eight out of twelve targets — without any further hyperparameter optimization. Interestingly, the achieved energy error lies significantly below the limit of 60 meV we mentioned in Sec. 3. This is likely due to two reasons. First, QM9 only contains small molecules, many of which are very rigid. These molecules do not have multiple conformers and their energy will thus be more deterministic. Second, QM9's data was generated by initializing

Table 4: MAE on coordinate-free QM9 (meV) for DimeNet$^{++}$ with synthetic coordinates. Both synthetic distances and angles yield significant improvements, together reducing the error by 24 % on average.

| | $\epsilon_{\text{HOMO}}$ | $U_0$ |
|---|---|---|
| No dist/angle | 74.1 | 41.9 |
| No angle (bounds+PPR) | 63.5 | 32.1 |
| distance & angle: | | |
| Bounds | 63.6 | 29.4 |
| PPR | 63.0 | 29.5 |
| Bounds+PPR | **61.7** | **28.7** |

Table 5: MAE on coordinate-free QM9. DimeNet$^{++}$ with synthetic coordinates outperforms previous models by 20 %, without any hyperparameter tuning. *Uses Mulliken partial charges.

| | Unit | GNN-FiLM* | GNN-FiLM+FA* | DimeNet$^{++}$-SC |
|---|---|---|---|---|
| $\mu$ | D | 0.238 | **0.226** | 0.303 |
| $\alpha$ | $a_0{}^3$ | 0.375 | 0.193 | **0.171** |
| $\epsilon_{\text{HOMO}}$ | meV | 52.5 | **47.7** | 61.7 |
| $\epsilon_{\text{LUMO}}$ | meV | 55.9 | **52** | 54.3 |
| $\Delta\epsilon$ | meV | 84.3 | **77** | 86.2 |
| $\langle R^2 \rangle$ | $a_0{}^2$ | 18.7 | 14.3 | **12.7** |
| ZPVE | meV | 13.2 | 5.62 | **2.98** |
| $U_0$ | meV | 233 | 68.8 | **28.7** |
| $U$ | meV | 256 | 75.2 | **29.6** |
| $H$ | meV | 240 | 83 | **29.6** |
| $G$ | meV | 222 | 76.1 | **28.2** |
| $c_\text{v}$ | $\frac{\text{cal}}{\text{mol K}}$ | 0.173 | 0.082 | **0.076** |

each molecule's position with a fast force field method. This can bias the final conformer towards a deterministic state, which might be learnable by a GNN.

**Synthetic coordinates without directional message passing.** In some cases we found that using synthetic coordinates yields performance improvements while transforming the model to a directional MPNN does not. Table 6 demonstrates this using DeeperGCN on QM9 and ogbg-molhiv. Using the line graph significantly impairs performance on both datasets. Whether directional MPNNs provide a benefit thus seems to depend on both the underlying model and the dataset. This is likely due to the directional MPNN's different training dynamics, which require further architectural and hyperparameter changes. Moreover, directional MPNNs

Table 6: Ablation of DeeperGCN on QM9 $U_0$ (MAE, meV) and ogbg-molhiv (ROC-AUC). Using the line graph does not always provide benefits. However, synthetic coordinates help even in these cases.

| | | QM9, $U_0$ | ogbg-molhiv |
|---|---|---|---|
| Basic | | 106 | $0.728 \pm 0.008$ |
| +distance | Bounds | 114 | $0.724 \pm 0.014$ |
| | PPR | **88** | $0.734 \pm 0.014$ |
| | Bounds+PPR | 100 | $0.733 \pm 0.024$ |
| +distance & line graph | Bounds | 233 | $0.705 \pm 0.011$ |
| | PPR | 204 | $0.697 \pm 0.009$ |
| +distance, line graph & angle | Bounds | 205 | $0.703 \pm 0.021$ |
| | PPR | 164 | $0.700 \pm 0.014$ |
| | Bounds+PPR | 186 | $\mathbf{0.767 \pm 0.016}$ |

are likely more prone to overfitting due to their better expressivity. This affects ogbg-molhiv in particular, since it uses a scaffold split for the test set. However, the additional information provided by synthetic coordinates still yields improvements in both cases.

# 7 Limitations and societal impact

**Limitations.** Converting a GNN to a directional MPNN incurs significant computational overhead, since the line graph is usually substantially larger than the molecular graph. However, just incorporating the information provided by graph distances or molecular distance bounds without transforming to directional message passing can also provide benefits, with almost no computational overhead. We furthermore found that transforming a GNN to a directional MPNN does not yield improvements in many cases, while synthetic coordinates still do (see Sec. 6 for details). There are likely also cases where synthetic coordinates lead to overfitting and do not improve accuracy. Directional MPNNs appear to be most successful when the molecular configuration is directly relevant.

**Societal impact.** Improving the predictions of molecular models can positively affect various applications in chemistry, biology, and medicine. Our research is general and not focused on a field where malicious use should be expected. However, similar to most methodological research, our improvements can be misused to accelerate the development of chemical agents and biological weapons. We do not think that this potential for harm goes beyond regular research in theoretical chemistry and related fields. Still, to slightly reduce these negative effects our code will be published under the Hippocratic license (Ehmke, 2020).

## 8  Conclusion

We proposed two methods for providing synthetic coordinates: Molecular distance bounds based on the interacting atom types, and graph-based distances based on personalized PageRank scores. Both of these methods provide well-defined pairwise distances, which can then be used to calculate distances for edge features in the molecular graph, and angles for edge features in its line graph. These synthetic coordinates improve GNN performance for various models and datasets, and allow transforming a regular GNN into a directional MPNN. This transformation leads to substantial improvements, resulting in state-of-the-art accuracies on multiple datasets.

## Acknowledgments and Disclosure of Funding

We would like to thank Johannes T. Margraf, Nicholas Gao, and Oleksandr Shchur for their invaluable advice and comments.

This research was supported by the Deutsche Forschungsgemeinschaft (DFG) through the Emmy Noether grant GU 1409/2-1 and the TUM International Graduate School of Science and Engineering (IGSSE), GSC 81.

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
