# A  Choosing hyperparameters

Table 7: ROC-AUC on ogbg-molhiv for various numbers of DeeperGCN layers. We use 12 layers.

| Layers | AUC-ROC |
|---|---|
| 7 | $0.754 \pm 0.028$ |
| **12** | $0.756 \pm 0.026$ |
| 15 | $0.742 \pm 0.028$ |

Table 8: ROC-AUC on ogbg-molhiv for various hidden layer sizes in DeeperGCN. We choose the largest, i.e. 256.

| Hidden size | AUC-ROC |
|---|---|
| 64 | $0.764 \pm 0.009$ |
| 128 | $0.760 \pm 0.026$ |
| **256** | $0.756 \pm 0.026$ |

Table 9: Different methods of embedding the angle for chemical distance bounds (ROC-AUC on ogbg-molhiv). Jointly using all 3 proposed components (Min+Max+Center) works best.

| Angle mode | MAE |
|---|---|
| Min | $0.754 \pm 0.048$ |
| Max | $0.754 \pm 0.013$ |
| Center | $0.744 \pm 0.012$ |
| Min+Max | $0.745 \pm 0.018$ |
| **Center+Min+Max** | $0.756 \pm 0.026$ |

Table 10: ROC-AUC on ogbg-molhiv. Different parameter sharing variants for the two layers used for embedding the distance and angle. Sharing the parameters of the first layer globally and using separate parameters per message passing step for the second layer ("Mixed") performs slightly better.

| Embedding method | MAE |
|---|---|
| Local | $0.754 \pm 0.010$ |
| Global | $0.753 \pm 0.025$ |
| **Mixed** | $0.756 \pm 0.026$ |

Table 11: Different bottleneck and basis sizes for embedding the distance and angle (ROC-AUC on ogbg-molhiv). We choose a 4-dimensional bottleneck, a 16-dimensional distance and a 18-dimensional angle embedding. Note that the latter numbers are the sum of all components, i.e. we use 8 dimensions for the minimum and 8 for the maximum distance.

| | Basis size | | |
|---|---|---|---|
| Bottleneck | Distance | Angle | MAE |
| | 4 | 6 | $\mathbf{0.762 \pm 0.021}$ |
| 2 | 8 | 9 | $\mathbf{0.766 \pm 0.019}$ |
| | 16 | 18 | $0.751 \pm 0.031$ |
| | 4 | 6 | $0.743 \pm 0.016$ |
| **4** | 8 | 9 | $0.756 \pm 0.026$ |
| | **16** | **18** | $\mathbf{0.767 \pm 0.016}$ |
| | 4 | 6 | $0.741 \pm 0.024$ |
| 8 | 8 | 9 | $\mathbf{0.771 \pm 0.015}$ |
| | 16 | 18 | $0.743 \pm 0.012$ |

In this section we highlight the best results as well as the chosen hyperparameters and model variants via bold font. We tune DeeperGCN on ogbg-molhiv to prevent selection bias and overfitting. Tables 7 and 8 show its performance for various choices of depth and width. Many of these results are not statistically significant. We chose a depth of 12 layers and a hidden size of 256.

Table 9 compares the three ways of representing the angle bounds described in Sec. 4. We see that simply using all three of them performs the best. Note that we keep the total basis size constant, i.e. we either use one 18-dimensional, two 9-dimensional, or three 6-dimensional angle bases.

We embed the information provided by synthetic coordinates using two linear layers with a small "bottleneck" layer in between and without non-linearities. Table 10 compares using separate local layers per message passing step against using a single global layer, i.e. sharing these parameters. Mixing these two variants by using one global and one local layer performs best. Table 11 furthermore compares different bottleneck and basis sizes for representing the distances and angles. A 4- or even 2-dimensional bottleneck performs best — which is surprisingly small compared to the used hidden size of 256. The basis size on the other hand is similar to the size used e.g. by Gasteiger et al. (2020b).