# OpenReview forum: "Directional Message Passing on Molecular Graphs via Synthetic Coordinates"
_NeurIPS.cc/2021/Conference — NeurIPS 2021 Poster_

### Official Review · Reviewer_FwTj · 2021-07-12

**Rating:** 6
**Confidence:** 4

**Summary:**

This paper aims to improve molecular property prediction by generating synthetic
3D coordinates. The method is based on the key observation that the true 3D
coordinates are difficult to obtain and oftentimes undecidable for molecular
structures. The paper proposes synthetic coordinates, which then enables using
GNNs without requiring the true molecular configuration. The experimental
results on several data sets have shown very promising results using this
method.


**Limitations And Societal Impact:**

Yes

**Main Review:**

This work generates synthetic 3D coordinates to improve molecular property
prediction. The key observation is that many molecule properties are related the
their 3d structures, but the true 3D coordinates are 3D coordinates are
difficult to obtain and oftentimes undecidable. The authors thus propose to
generate "synthetic" coordinates, which enables using GNNs without requiring the
true molecular configurations.

The synthetic coordinates provide pairwise distances between atoms, which can
then be used to calculate distances and angles as edge features for the 2D
molecule graphs. This can be then used to enhance GNNs into directional MPNNs to
improve their predictive power. The authors propose two methods for generating
synthetic coordinates: 1) graph-based distances based on personalized PageRank
scores; and 2) molecular distance bounds based on the interacting atom types.

The overall idea of incorporating synthetic coordinates into GNNs is well
motivated, and has shown very good performance. However, the design of the two
proposed methods for obtaining synthetic coordinates are quite arbitrary and
heuristic. This is especially true for the personalized pagerank method for
generating distances. The authors first calculate PPR scores and then turn that
into distances using the arccos function. But why does this conversion makes
sense? How can the arccos of PPR scores approximate the true 3D distance?

The experimental results are quite positive. The author have shown that such
"synthetic" coordinates enhancements can improve the performance of regular GNNs
on multiple benchmarks.



**Time Spent Reviewing:**

4 hours

---

> ### Author Response · Authors · 2021-08-09
> **Principled synthetic coordinates**
>
> We are happy to hear that you find our idea well-motivated and the shown performance very good!
>
> ### Principled choice of coordinates
>
> While we understand that PPR may look unfamiliar to molecular modeling researchers, it is actually a well-motivated graph-theoretical concept. We have added several explanations to better introduce PPR in the paper (see response to reviewer u3W7). The way we transform PPR to a distance is the simplest way of computing a metric from a positive semi-definite kernel. We first slightly alter the PPR score to obtain a symmetric kernel. We then compute the metric via
> $$
> d_{\text{sppr},ij} = \sqrt{\Pi_{ii}^{\text{sppr}} + \Pi_{jj}^{\text{sppr}} - 2\Pi_{ij}^{\text{sppr}}},
> $$
> which is just the induced distance in the kernel's reproducing kernel Hilbert space (as described in lines 177-180). We then compute the angle with standard trigonometry by the three lengths of a triangle, using the arccos (Eq. 6). We would like to explicitly highlight that the **arccos is not used in the distance calculation**.
>
> ### PPR is not related to 3D distance
>
> As we note in lines 180-181, the PPR distance is not related to the true 3D distance. We have extended this statement to stress this point further:
> > However, $d_{\text{sppr},ij}$ is a general metric and does \emph{not} yield atom positions in 3D. This is a purely graph-based measure that does not incorporate any chemical knowledge. It reflects how central an atom is in the molecular graph, and how important another atom is to this one, based on the overall network of bonds. It thus only helps the GNN better reflect and process the molecular graph structure.
> We have furthermore added a figure showing an example of $d_{sppr}$ on ethanol.

---

### Official Review · Reviewer_4tVD · 2021-07-16

**Rating:** 4
**Confidence:** 4

**Summary:**

The paper proposes a method for estimating molecular properties without 3D information about the molecules. The main contribution of the paper is an approach to approximate the local distance and angular information. This may then be fed into other GNNs for calculating the various properties. The authors also explore improvements related to using line graphs.

Results are shown on QM9, ZINC and ogbg-molhiv datasets.


**Ethical Concerns:**

None.

**Limitations And Societal Impact:**

Good.

**Main Review:**

The authors propose an interesting approach to approximating local geometry information in molecules. It is a novel and potentially useful technique when geometry information cannot be obtained, as the authors describe in Section 3. However, the authors never compare against other techniques for computing approximate geometry information, such as force fields or direct approaches. Without these experiments the paper is below the acceptance threshold.

\+ Interesting approach that sidesteps the need for 3D geometry information when predicting molecular properties.

\+ Initial results look promising.

\-- No comparison with other approximate 3D geometry estimation techniques. Comparison should be done with force field methods or direct methods (i.e., other computationally efficient approaches). Otherwise the usefulness of the author’s technique is unknown.

\- ML models are also trained to predict atomic forces (SchNet, DimeNet, etc.). These may then be used to predict relaxed structures for the molecules (potentially finding multiple conformers). These methods are also computationally very fast and may be more accurate than force fields, especially for small molecules.

\- The authors state that conformers are a significant problem. However, the models can be run on multiple potential conformers and the ones with lowest energy selected. This won’t work if the energy isn’t estimated, but may be true for many applications.

\- On line 193, the authors state “The angles in ring systems are handled separately based on a set of hand-crafted rules.” It’d be better is a principled method was used in all cases.

\- The results in Table 5 contradict the earlier results. It would greatly strengthen the paper if the discrepancy was explained.

Other comments:

* Move limitations after the results section. This will give the reader better context on the discussion.

* Recommend not emphasizing the 49% improvement with DeeperGCN since it’s not the SOTA approach (Table 1). Better to state the improvement on the best method, SMP.

* The labels in the tables are confusing as to which settings are used. E.g., in Table three, do PPR and bounds use angles (I assumed so)?


**Time Spent Reviewing:**

2.5 hours

---

> ### Author Response · Authors · 2021-08-09
> **Comparison to other approximate 3D geometry methods; conformer search, improved explanations**
>
> We are happy to hear that you find our approach interesting, novel and potentially useful, and our results promising!
>
> ### New results for approximate geometry computation
> We already show a result using force fields (MMFF94) in line 308. We have now run additional experiments for force fields (using MMFF94) and direct configuration prediction methods (using ETKDG) on ZINC with DeeperGCN. Note that in the meantime we have run all the paper's experiments with 5 repetitions to make them more robust and additionally report the standard deviation. This is why these numbers vary slightly from the paper.
>
> |                               |     MMFF94     |     ETKDG      |       Bounds       |        PPR         |     Bounds+PPR     |
> |-------------------------------|----------------|----------------|--------------------|--------------------|--------------------|
> | distance                      | 0.299 +- 0.021 | 0.284 +- 0.012 |   0.304 +- 0.022   | **0.282** +- 0.009 |         -          |
> | distance + line graph         | 0.228 +- 0.007 | 0.223 +- 0.008 | **0.218** +- 0.007 |   0.223 +- 0.004   |         -          |
> | distance + line graph + angle | 0.225 +- 0.007 | 0.225 +- 0.008 |   0.197 +- 0.010   |   0.178 +- 0.010   | **0.161** +- 0.007 |
>
> We can see that our simpler, faster, and more principled methods (bounds, PPR) actually perform better than more sophisticated coordinate prediction methods, as stated in the paper.
>
> ### ML-based conformer search
>
> Using ML-based potentials to find an (ensemble of) accurate conformers and using this for prediction is certainly a very interesting approach. Unfortunately, obtaining an unbiased ensemble of conformers requires expensive post-processing, defeating the purpose of fast ML methods (as discussed in lines 162-167). Together with the negative results from other accurate conformer generation methods (see above), this motivates us to focus on very simple and fast methods instead.
>
> ### Using multiple conformers
>
> We agree that this approach is very promising, but much more challenging than it may seem. If we optimize a structure with an ML potential we overfit on this potential (by definition), so the true energy will be severely underestimated. Selecting the conformer thus requires a second, independent model. Obtaining two independent, highly accurate models seems quite hard, both from a modelling and training perspective; and using quantum-mechanical methods would again defeat the purpose of scalable ML-based methods. Moreover, finding the true lowest-energy conformer is likely extremely hard, considering the complexity of this non-convex optimization problem. Still, we do think this is a very promising approach for future work. We consider our synthetic coordinates merely as one simple but effective method for using on-the-fly 3D structures to boost model performance.
>
> ### Ring systems
>
> We have changed lines 192-193 to:
> > The angles between triplets of atoms are estimated based on bond hybridization and whether an atom is part of a ring. The distance bounds between two-hop neighbors are then calculated based on this angle, the bond length, and a tolerance of 0.04A, or 0.08A for atoms larger than Aluminium.
>
> Note that the bounds matrix we use is based on well-founded principles, and used extensively throughout the popular RDKit library.
>
> ### Table 5
>
> We have extended the existing discussion in lines 336-342 with
> > This is likely due to these models' different training dynamics, which require further architectural and hyperparameter changes. Moreover, directional MPNNs are likely more prone to overfitting due to their better expressivity. This affects ogbg-molhiv in particular, since it uses a scaffold split for the test set.
>
> Note that it is our conviction that science always benefits from the full picture, including its complicated and inconsistent sides. This is why we have also included Table 5. We hope that the reviewer can appreciate this as well.
>
> ### Minor comments
>
> Thank you for the additional comments! We moved the limitations back and added a line noting "distance+angle" in Table 3. We mention the result on DeeperGCN as well since it is a rather basic GNN, basically GCN with some architectural improvements. SMP performs very well, but it is a rather advanced model. Showing good improvements with a more "standard" GNN seems valuable to us.

---

### Official Review · Reviewer_u3W7 · 2021-07-17

**Rating:** 6
**Confidence:** 3

**Summary:**

This paper focuses on the task of molecular property prediction and proposes novel approaches to build synthetic coordinates (geometric information) in molecules when atom positions are not available for advanced GNNs. Since the performance of those GNNs highly relies on atom coordinates, the applicable cases of advanced GNNs are limited without reliable and efficient approaches to generate coordinates before training.

The authors propose two ways to generate the synthetic coordinates: 1. Graph-based distance and angles via Personalized Page-Rank; 2. Molecular distance bounds and corresponding angles based on chemical knowledge. To embed the resulting distances and angles, the authors propose to convert the original molecular graphs to line graphs. With line graphs as input, regular GNNs work as directional MPNNs, which exhibit better performance in practice. Experiments on benchmarks validate the improvements achieved by using synthetic coordinates in molecular graphs with several baseline models.

**Limitations And Societal Impact:**

The authors have addressed the limitations and potential negative social impact of their work in section 5.

**Main Review:**

The proposed synthetic coordinates address the gap between the performance of GNNs on molecular property predictions when the atom coordinates are provided and not provided. The design of advanced GNNs with available atom positions is also different from regular GNNs when only molecular graphs are achievable. With synthetic coordinates and line graphs, regular GNNs can be converted to directional MPNNs, which have the potential to gain better performance with distance and angles as features. The proposed idea is novel and contributes to the related field especially molecular property prediction is an important subarea now.

Overall the paper is well-written but can be better organized: I feel that the "Molecular configurations" section is a bit too long for discussing the background chemical knowledge and motivations. In contrast, the "Synthetic coordinates" section that describes the proposed methods is too concise without touching enough details. Some other suggestions for "Synthetic coordinates" section:

(1) The proposed methods can be explained with values computed on certain example molecules for better understanding.

(2) It would be interesting to compare the synthetic coordinates achieved by the two methods to find any possible relations.

(3) As for the implementation, although the authors mention combining both two synthetic coordinates, it is not clear how the two coordinates are implemented together in GNNs.

The results in the experiment section clearly demonstrate the improvements achieved by introducing synthetic coordinates and line graphs. I would suggest that Table 1 and Table 5 can be better written since it seems that each ablated model only contains one modification (distance, line graph, or angle), while "+angle" actually means "+disrance & line graph & angle". By only reading the tables, it is unknown that the changes are done step-by-step.



**Time Spent Reviewing:**

2

---

> ### Author Response · Authors · 2021-08-09
> **More well-organized paper, revised Section 4**
>
> We are glad that you find our idea novel, the target problem important, our paper well-written, and that our experiments clearly demonstrate the achieved improvements. We have incorporated all of your comments, leading to a **more well-organized and readable paper**.
>
> ### Improved Section 3 (molecular configurations)
>
> As suggested, we have shortened this section, e.g. by removing the details of the MMFF94 force field. Note that while the discussion on conformers section might seem excessive, we felt it necessary to properly introduce an ML practitioner to this topic, discuss the associated problems, and motivate our method. As evidenced by reviewer oh4Z's initial misunderstanding, this is section is important to make sure that people fully understand our approach.
>
> ### Revised Section 4 (synthetic coordinates)
>
> Based on your feedback we have extensively improved the "Synthetic coordinates" section:
> - added an explanation of PPR (in line 172):
> > PPR measures how close two atoms in the molecular graph are by calculating the probability that a random walker starting at atom $i$ ends up at atom $j$. At each step, the random walker jumps to any neighbor of the current atom with equal probability, and teleports back to the original atom $i$ with probability $\alpha$.
> - added an explanation of the PPR distance, including an example figure, as suggested (in line 181, Fig. 1 already contains an example with bond distances)
> > This is a purely graph-based measure that does not incorporate any chemical knowledge. It reflects how central an atom is in the molecular graph, and how important another atom is to this one, based on the overall network of bonds. It thus only helps the GNN better reflect and process the molecular graph structure. Fig. 2 shows an example of $d_{\text{sppr}}$ on ethanol.
> - more comprehensively specified how we obtain distance bounds for two-hop neighbors (instead of lines 192-193):
> > The angles between triplets of atoms are estimated based on bond hybridization and whether an atom is part of a ring. The distance bounds between two-hop neighbors are then calculated based on this angle, the bond length, and a tolerance of \SI{0.04}{\angstrom} or \SI{0.08}{\angstrom} for atoms larger than Aluminium.
> - revised lines 207-213 to give a more complete explanation of how to incorporate synthetic coordinates (note that we have already explained further details in the appendix):
> > **Representing distances and angles.** The additional structural information can directly be incorporated into existing models as edge features. For this purpose, we propose to first represent the distances using $N_\text{RBF}$ Gaussian radial basis functions (RBF), i.e.
> > $$h_{\text{RBF}, n}(d_{ij}) = \exp^{-1/2 (d_{ij} - c_n)^2 / \sigma^2},$$
> > where the Gaussian centers $c_n$ are set uniformly between 0 and the maximum distance, $n \in [0, N_\text{RBF}]$, and $\sigma = c_1 - c_0$ is set as the distance between two neighboring centers. The angles are similarly represented using $N_\text{ABF}$ cosine angular basis functions (ABF), i.e.
> > $$h_{\text{ABF}, n}(\alpha_{ijk}) = \cos(n \alpha_{ijk}),$$
> > with $n \in [0, N_\text{ABF}]$. We then transform these features using two linear layers. The first layer is global and uses a small output dimension to force the model to learn a well-generalizing intermediate representation. The second layer is specific to each GNN layer, enabling more flexibility. Overall, we obtain the distance-based edge features $e_{ij}$ and angle-based triplet features  $a_{ijk}$ in layer $l$ via
> > $$e_{ij}^{(l)} = W_{\text{RBF}2}^{(l)}W_{\text{RBF}1} (h_{\text{RBF}}(d_{ij}) \| x_{ij}^{(\mathcal{E})}),$$
> > $$a_{ijk}^{(l)} = W_{\text{ABF}2}^{(l)}W_{\text{ABF}1} h_{\text{ABF}}(\alpha_{ijk}),$$
> > where $W_{\text{RBF}2}^{(l)}$ and $W_{\text{ABF}2}^{(l)}$ are layer-wise learned weight matrices, $W_{\text{RBF}1}$ and $W_{\text{ABF}1}$ are global learned weight matrices, $\|$ denotes concatenation, and $x_{ij}^{(\mathcal{E})}$ are bond (edge) features. We can furthermore combine multiple synthetic coordinates by concatenating their representations $h_{\text{RBF}}$ or $h_{\text{ABF}}$.
>
> ### Improved Tables 1 & 5
>
> As suggested, we have made the tables clearer by changing the first column to "+ distance + line graph" and "+ distance + line graph + angle".
>
> Thank you for the invaluable feedback! We are happy to incorporate further improvements and hope that these revisions have updated your opinion.

---

### Official Review · Reviewer_oh4Z · 2021-07-20

**Rating:** 3
**Confidence:** 5

**Summary:**

The authors use a generic algorithm to embed molecular graph into 3D space using a distance matrix computed from PageRank. Then they use this 3D information as additional features to do property prediction. Results show that a) this approach is better than using GNNs without this information b) the augmented structures can even be applied to on Euclidean GNNs such as DimNet and get better results compared to simple GNNs.

**Limitations And Societal Impact:**

Yes

**Main Review:**

This paper attempts to construct a learning approach that is not meaningful for molecular setting. Specifically, the authors are trying to predict properties that are a function of chemical graph *and* configuration (3D info) from the chemical graph alone. This is done by adding pseudocoordinate information extracted from the chemical graph itself. However, since no real coordinate information is added this just seems to indicate that some inefficiency in the GNNs compared to was exploited - it must be possible to get similar performance with suitable GNNs because they use exactly the same input information eventually.

More fundamentally, it is simply meaningless to predict a property of the 3d structure independent of the 3d structure. What *would* be meaningful is to predict an ensemble average over 3d structures such as a solvation free energy as a function of the chemical graph alone, but the prediction of the QM9 properties certainly isn't. As a result, I am not convinced of any of the results.

**Time Spent Reviewing:**

4

---

> ### Author Response · Authors · 2021-08-09
> **Conformers, distance bounds, predictive limits, and other datasets**
>
> We suspect that your preliminary review is largely based on an **unfortunate misunderstanding**. We are aware of the mentioned issues and discuss them extensively in the paper:
> 1. We discuss the issue associated with the ambiguity of conformers at length in Section 3. We also discuss the best accuracy we can expect without knowing the exact coordinates and motivate our method from this, based on multiple recent papers on conformer search (lines 128-133 and 157-167).
> 2. Based on this discussion, we propose to use **distance bounds, not exact 3D structures**. By doing so, we effectively represent the molecule by ranges of interatomic distances. These ranges include all chemically reasonable 3D structures. This provides valuable chemical and structural information for the model.
> 3. The PPR distance is a _second_ piece of information that we propose. This has no real connection to the actual 3D structures (as highlighted in lines 180-181). It only conveys additional information of the molecular graph to the model, thus improving GNN performance. A GNN might in principle be able to learn this information. However, adding it directly makes the task easier and leads to better performance and sample efficiency. We thus indeed "exploit some inefficiency in the GNNs", and fix the associated problem with a simple and fast method. Note that we **use best-in-class GNNs** for our experiments, not cherry-picked examples. We are able to improve these and set the state of the art on multiple datasets. This is thus currently the best known way of alleviating these "inefficiencies". We have added this statement to Section 4 (in line 181) to clarify this further:
> > However, $d_{\text{sppr},ij}$ is a general metric and does \emph{not} yield atom positions in 3D. This is a purely graph-based measure that does not incorporate any chemical knowledge. It reflects how central an atom is in the molecular graph, and how important another atom is to this one, based on the overall network of bonds. It thus only helps the GNN better reflect and process the molecular graph structure.
> 4. As discussed in the paper, estimating the properties of QM9 can make sense up to a certain accuracy, since there are only a few reasonable equilibrium structures (conformers) for small molecules, and QM9 is limited to equilibrium structures. The model thus effectively needs to estimate a likely equilibrium structure along with the target property. But there are fundamental limits to this and we cannot expect nearly the same accuracy as when knowing the 3D structure. This approach effectively yields **an estimate, not a full prediction**. Again, we discuss this in Section 3. We further discuss the unexpectedly good results on QM9 in the experimental section (lines 322-327). Additionally, we have added this statement to the experimental setup to stress this even more:
> > Note that the properties in QM9 depend on the molecule's precise configuration. The predictions in coordinate-free QM9 should thus be viewed as estimates for the equilibrium configurations. There are fundamental limits to the accuracy achievable in this setup, as discussed in Section 3.
> 5. We evaluate on 3 datasets, and QM9 is just one of them. The other 2 are concerned with more high-level properties: In ZINC we predict the constrained solubility and on ogbg-molhiv we predict whether a molecule inhibits HIV virus replication. Both these properties depend on the full ensemble of conformers, and not on one specific 3D structure.
>
> In summary, we discuss the mentioned issues extensively in the paper, propose a method directly motivated by these considerations, discuss the limitations of our estimates on QM9, and evaluate on additional datasets and targets.
>
> We hope that our response and changes have made this more clear. We are happy to extend this discussion further if the reviewer feels that we missed another specific aspect.

---

### Author Response · Authors · 2021-08-09
**Improved paper structure and explanations, comparison with conformer generation methods,**

We would like to thank all reviewers for their diligent effort and invaluable feedback! We are happy to hear that you find our method novel and the experimental results very positive! We have significantly improved the paper structure and explanations based on your feedback, and added an important comparison with conformer generation methods. We hope that we were able to clarify any doubts, and are more than happy to answer any further questions.

Additionally, we have independently repeated each experiment five times in the meantime to make sure that the results are statistically robust and provide standard deviations. The new results are in line with or better than in the submitted version of the paper. For example, on ZINC we now obtain an MAE of 0.131+-0.004 for SMP-SC, and 0.161+-0.007 for DeeperGCN-SC.

---

### Decision · Program_Chairs · 2021-09-27

**Decision:**

Accept (Poster)

**Comment:**

The ratings were 3, 6, 4, 6.

The reviewer who gave the 3 wrote "it is simply meaningless to predict a property of the 3d structure independent of the 3d structure", however the authors gave a detailed response that suggests there might have been some confusion, and the reviewer didn't follow up. Moreover, this approach effectively uses carefully chosen inductive biases, which seems to go beyond "some inefficiency in the GNNs compared to was exploited" as the reviewer suggests.

For the reviewer who gave the 4, since the authors provide some new results in response, the score probably should be elevated.

While this is still a borderline paper, I recommend "accept" given that the results are pretty good, there doesn't seem to be any technical problem, and there is clear novelty.